# Association between Skeletal Muscle Loss and the Response to Neoadjuvant Chemotherapy for Breast Cancer

**DOI:** 10.3390/cancers13081806

**Published:** 2021-04-09

**Authors:** Byung Min Lee, Yeona Cho, Jun Won Kim, Sung Gwe Ahn, Jee Hung Kim, Hei Cheul Jeung, Joon Jeong, Ik Jae Lee

**Affiliations:** 1Department of Radiation Oncology, Yonsei University College of Medicine, Seoul 03722, Korea; bmlee9125@yuhs.ac (B.M.L.); iamyona@yuhs.ac (Y.C.); junwon@yuhs.ac (J.W.K.); 2Department of Surgery, Yonsei University College of Medicine, Seoul 03722, Korea; asg2004@yuhs.ac; 3Division of Medical Oncology, Yonsei University College of Medicine, Seoul 03722, Korea; ok8504@yuhs.ac (J.H.K.); jeunghc1123@yuhs.ac (H.C.J.)

**Keywords:** breast cancer, skeletal muscle loss, treatment-induced cachexia

## Abstract

**Simple Summary:**

The loss of skeletal muscle mass is known to be associated with poor treatment outcome, treatment-related toxicity, and high mortality. The association between loss of skeletal muscle mass and the response to treatment is not well-defined yet. In this study, we evaluated the impact of loss of skeletal muscle mass on responsiveness to neoadjuvant chemotherapy in breast cancer. The prediction of response to neoadjuvant chemotherapy could be helpful to guide the treatment direction.

**Abstract:**

There are no means to predict patient response to neoadjuvant chemotherapy (NAC); the impact of skeletal muscle loss on the response to NAC remains undefined. We investigated the association between response to chemotherapy and skeletal muscle loss in breast cancer patients. Patients diagnosed with invasive breast cancer who were treated with NAC, surgery, and radiotherapy were analyzed. We quantified skeletal muscle loss using pre-NAC and post-NAC computed tomography scans. The response to treatment was determined using the Response Evaluation Criteria in Solid Tumors. We included 246 patients in this study (median follow-up, 28.85 months). The median age was 48 years old (interquartile range 42–54) and 115 patients were less than 48 years old (46.7%). Patients showing a complete or partial response were categorized into the responder group (208 patients); the rest were categorized into the non-responder group (38 patients). The skeletal muscle mass cut-off value was determined using a receiver operating characteristic curve; it showed areas under the curve of 0.732 and 0.885 for the pre-NAC and post-NAC skeletal muscle index (*p* < 0.001 for both), respectively. Skeletal muscle loss and cancer stage were significantly associated with poor response to NAC in locally advanced breast cancer patients. Accurately measuring muscle loss to guide treatment and delaying muscle loss through various interventions would help enhance the response to NAC and improve clinical outcomes.

## 1. Introduction

Neoadjuvant chemotherapy (NAC) for breast cancer has a high clinical response rate. In some cases, NAC facilitates the conversion of unresectable, locally advanced breast cancer to operable cancer [1,2]. Consequently, conversion to resectable status leads to an increase in the breast conservation rate [3,4]. Some patients undergoing NAC for breast cancer show an insufficient response to therapy. Currently, we do not have the means to predict whether patients will respond well to NAC or not.

The loss of skeletal muscle mass is known to be a poor prognostic factor in patients with malignancies. It is highly associated with chemotherapy toxicity [5,6], tumor progression [6], and mortality [7]. The impact of skeletal muscle loss on the response to treatment remains undefined. Although there is no established standard method for skeletal muscle quantification, computed tomography (CT) is used to measure the skeletal muscle mass. CT images are frequently obtained for cancer patients as a part of staging and evaluating the progress of disease during treatment.

To our knowledge, this study is the first to investigate the association between skeletal muscle loss and responsiveness to NAC. In this study, we analyzed the association between skeletal muscle loss and the response to NAC to evaluate the impact of skeletal muscle loss in patients receiving NAC for locally advanced breast cancer. Being able to predict the response to NAC would help guide the treatment direction.

## 2. Patients and Methods

### 2.1. Patient Population

Patients diagnosed with breast cancer who were treated with NAC, surgery, and radiotherapy (RT) from March 2013 to September 2019 were included and analyzed. The chemotherapy regimen and schedule were decided at the physician’s discretion. The clinical and laboratory data of patients were collected through electronic medical records.

We analyzed 246 patients in total. The inclusion criteria for the patients were as follows: (1) age more than 18 years, (2) treatment with either partial mastectomy or total mastectomy, (3) treatment with NAC and adjuvant RT, and (4) available medical records. The exclusion criteria for patients were as follows: (1) follow-up loss, (2) inability to measure the skeletal muscle mass, and (3) synchronous malignancies. Every patient in this study received neoadjuvant chemotherapy, operation, and adjuvant radiotherapy. The adjuvant radiotherapy was performed in patients with tumor stage III/IV or extensively positive lymph node initially among the patients who underwent total mastectomy. This study was approved by the institutional review board of Gangnam Severance Hospital (3-2020-0509). Because the study was retrospective, the need for written informed consent was waived.

### 2.2. Measurement of Body Composition and the Definition of Skeletal Muscle Loss

CT was used to measure body composition. Generally, skeletal muscle mass is evaluated at the level of the third lumbar vertebra (L3) on abdominal/pelvic CT scans. Most of the patients in our study underwent abdominal/pelvic CT for the assessment of metastatic lesions. However, a few patients did not undergo abdominal/pelvic CT. Some recent studies have evaluated the skeletal muscle index (SMI) at the level of the fourth thoracic vertebra (T4) on chest CT scans [8,9]. Similarly, we assessed the SMI at the upper border of T4 (Figure 1).

We estimated the SMI using CT scans taken within 3 months of diagnosis and those taken for radiotherapy (RT) simulations. CT was performed before the initiation of NAC and after the completion of NAC; thus, pre-NAC and post-NAC CT scans were obtained for each patient. We used the MIM Vista software (MIM corp., Version 6.6.14., Cleveland, OH, USA) to delineate the body composition based on Hounsfield units (HUs). The following HU threshold was applied for skeletal muscle delineation: from −29 to +150. The cross-sectional volumes obtained from the MIM software at the T4 level were divided by the thickness of the axial slice and defined as the cross-sectional areas. The SMI was calculated by dividing the cross-sectional areas by the height of the patients.

Similar to skeletal muscle, subcutaneous adipose tissue composition was delineated based on HUs: −190 to −30 HU. The subcutaneous adipose tissue index was determined by dividing the cross sectional volume of subcutaneous adipose tissue at the T4 level with thickness of axial slice.

### 2.3. Statistical Analysis

We aimed to investigate the association between skeletal muscle loss status and responsiveness to treatment. Responsiveness to treatment was evaluated according to the Response Evaluation Criteria in Solid Tumors [10]. The treatment response was evaluated based on the image study of the patients. Magnetic resonance imaging (MRI) was used for the evaluation before and after NAC. Patients showing either a complete response or partial response in the MRI performed after NAC were classified into the responder group, while patients with either stable disease or progressive disease were categorized into the non-responder group.

For categorical data, Fisher’s exact test was used for comparison between the groups. The Mann–Whitney U-test was used to compare continuous variables between the groups. The logistic regression method was used to determine the association between the variables and the response to NAC. A *p*-value < 0.05 was considered statistically significant. A multivariate analysis was conducted among the significant variables in the univariate analysis, with the calculation of hazard ratios and 95% confidence intervals. IBM SPSS version 25.0 (SPSS, Chicago, IL, USA) was used for analysis.

## 3. Results

### 3.1. Patient and Tumor Characteristics

We included 246 patients with breast cancer in this study. The median follow-up duration was 28.85 months (interquartile range [IQR], 18.20–41.10 months). The median age was 48 years (IQR, 42–54 years). Thirty-one patients had hypertension (12.6%), and 17 patients had diabetes mellitus (6.90%). Among the entire cohort, 71 patients performed regular exercise (28.9%). During the treatment, 21 patients showed weight gain (8.5%), while 14 patients showed weight loss (5.7%). Thirty patients had skeletal muscle loss before NAC (12.20%), while 33 patients were stratified as having skeletal muscle loss based on the post-NAC CT scan (Table 1). Among the 30 patients who showed skeletal muscle loss on the pre-NAC CT scan, 18 showed skeletal muscle loss both on the pre-NAC and post-NAC CT scans (7.3%), while 12 patients did not show skeletal muscle loss on the post-NAC CT scan (4.9%). Fifty patients showed a transition to skeletal muscle loss status after NAC (20.3%). Also, the body mass index (BMI) of the patients was evaluated. Eight patients showed underweight, with BMI less than 18.5 (3.3%), while 43 patients stratified as overweight (17.5%).

Tumor characteristics are presented in Table 2. The rates of estrogen receptor (ER) and progesterone receptor (PR) positivity were 59.8% and 30.5%, respectively, and 30.9% of patients tested positive for human epidermal growth factor receptor 2 (HER2). Sixty-one patients had stage T3/4 tumors (24.8%), and 114 patients had stage III tumors (46.3%). Breast-conserving surgery was performed in 129 patients (52.4%). In terms of adjuvant treatment, 103 patients (41.9%) received a selective ER modulator, and 26 patients (10.6%) were treated with an aromatase inhibitor. Trastuzumab-based NAC was administered to 72 patients (29.3%).

### 3.2. Cut-Off Value of the SMI for Skeletal Muscle Loss Group

We used the receiver operating characteristic (ROC) curve to determine the optimal cut-off value of SMI at the T4 level as there is no established value. The SMI value of 6, both pre-NAC and post-NAC, showed good sensitivity and specificity for responsiveness to NAC. On the basis of the area under the ROC curve (AUC) of 0.732 (Figure 2), we identified the pre-NAC SMI at the T4 level as a good predictor of responsiveness to NAC. Figure 3 demonstrates the ROC curve of the post-NAC SMI for predicting responsiveness to NAC; the AUC was 0.885, indicating that it was as an excellent predictor.

### 3.3. Predictive Value of Subcutaneous Adipose Tissue Index

We also analyzed the subcutaneous adipose tissue index (SATI) to evaluate whether the SATI can predict the response to NAC. Both the pre-NAC and post-NAC SATI showed poor specificity and sensitivity in predicting responsiveness to NAC (Appendix A). The AUCs of pre-NAC and post-NAC SATI were 0.560 and 0.587, respectively, indicating SATI to be a poor predictive tool.

### 3.4. Comparison of Patient and Tumor Characteristics Between the Responder and Non-Responder Groups

Among the 246 patients, 208 patients responded well to NAC, while 38 patients showed a poor response. Most of the variables were well balanced between the responder and non-responder groups. Skeletal muscle loss at both the pre-NAC and post-NAC were associated with responsiveness to NAC. The non-responder group included more patients with skeletal muscle loss (non-responder group vs. responder group: pre-NAC skeletal muscle loss patients; 23.7% vs. 10.1%, *p* = 0.019, post-NAC skeletal muscle loss patients; 55.3% vs. 5.8%, *p* < 0.001) (Table 3). The non-responder group had more features of tumor aggressiveness, such as advanced stage and a high Ki-67 index (responder group vs. non-responder group: stage III, 41.8% vs. 71.10%, *p* = 0.003; Ki-67 index ≥ 15%, 28.8% vs. 47.4%, *p* = 0.001) (Table 4).

### 3.5. Univariate and Multivariate Analyses of the Response to Neoadjuvant Chemotherapy

In the multivariate Cox proportional hazard models, post-NAC skeletal muscle loss was significantly associated with the response to NAC (Table 5). Pre-NAC skeletal muscle loss could also significantly predict the response to NAC. Along with skeletal muscle loss, advanced disease stage was associated with a significantly poorer response to NAC (hazard ratio 0.28, 95% confidence interval 0.11–0.71, *p* = 0.007). 

## 4. Discussion

To our knowledge, this is the first study to evaluate the association between skeletal muscle loss and the response to NAC in patients with locally advanced breast cancer. Post-NAC skeletal muscle loss is more strongly related to responsiveness to NAC than pre-NAC skeletal muscle loss. We also classified the adiposity value to evaluate its effect on the response to NAC, but there was no association.

CT-determined skeletal muscle mass is usually evaluated at the level of the L3 spine. We used the SMI at the T4 level for easy evaluation as all patients with breast cancer underwent chest CT before the start and after the completion of NAC. There is no defined cut-off value for the SMI at the T4 level to evaluate skeletal muscle mass. For this reason, the value determined by the ROC curve was set as the cut-off value in this study. Further studies are needed to establish the optimal cut-off value of the SMI at the T4 level for skeletal muscle loss.

In our study, skeletal muscle loss was significantly associated with the response to NAC. There are several approaches to predicting the response to NAC. Physicians have been trying to predict the effect of NAC using imaging tools [11,12] and biomarkers [13,14]. However, the obvious significant factors related to the response to NAC have not yet been defined. This may be due to the diverse clinical outcomes depending on the molecular subtypes of breast cancer [15]. We demonstrated that skeletal muscle loss is one of the predictive markers for the response to NAC. Along with skeletal muscle loss, the clinical stage of the cancer can predict the response to NAC. This is concordant with previous studies showing that the clinical stage is associated with the response to NAC [16,17,18]. More advanced stage or a larger tumor size are related to a poorer response to NAC, indicating that a higher tumor burden may have a negative impact on the response to treatment.

Obesity is regarded as a poor prognostic factor for breast cancer [19,20]. Patients with a high body mass index (BMI) have been reported to achieve fewer pathologic complete responses after NAC [21,22]. This would indicate the significance of BMI as a predictive factor for the response to NAC. However, there was no relation between high adiposity and the response to NAC in this study. There can be two reasons for this. First, the assessment of obesity differed according to studies. We assessed obesity based on CT-determined SATI, while other studies evaluated obesity using BMI. The results could vary because of differences in measurement methods. Another possible reason is the racial difference. The number of people with a high BMI is lower in Asian populations than in Caucasian populations, and the proportion of patients with obesity was much lower in Asians [23]. Our study could be inadequate to demonstrate an association between obesity and the response to NAC as the number of patients with obesity was inadequate.

Recent studies have demonstrated that a low level of muscle mass is highly associated with inflammation markers [24,25]. The local immune response drives muscle breakdown and, consequently, leads to systemic inflammation and tumor growth [26]. Indicators of inflammation were well-correlated with the response to treatment in various malignancies [27,28]. One study reported that inflammation is a key factor for chemotherapy-induced skeletal muscle loss in patients with cancer [29]. The levels of inflammatory cytokines, such as interleukin 8, tumor necrosis factor alpha, and C-reactive protein, were increased in patients with treatment-induced skeletal muscle loss, while no changes occurred in patients without treatment-induced skeletal muscle loss. Thus, inflammation might play a key role in enhancing treatment-induced skeletal muscle loss and affecting the response to NAC. Several studies reported that skeletal muscle loss is associated with chemotherapy toxicity [30,31]. Besides, one study reported that skeletal muscle loss is associated with intolerance to treatment [32]. Prospective clinical trials are needed to further understand the relation between the loss of muscle mass, inflammation, and the response to NAC as well as the mechanisms underlying these relationships.

There are several limitations to this study. Due to the retrospective nature, the causal relationships between the variables and skeletal muscle loss could not be determined. The results of this study indicated that muscle wasting at the end of neoadjuvant treatment is more associated with the response to treatment. Muscle wasting during treatment could be influenced by tumor aggressiveness. Due to this reason, careful attention should be paid in interpreting the results. Furthermore, data for some of the variables were unavailable. For example, we could not evaluate inflammatory status based on the levels of C-reactive protein, an objective marker for inflammatory status. Further, due to the short follow-up, we could not evaluate survival parameters such as overall survival and progression-free survival. Lastly, the cut-off value of the SMI was not a confirmed optimal cut-off value. Further studies are needed to determine the optimal cut-off value for CT-determined skeletal muscle loss. Despite these limitations, this is the first study demonstrating the relationship between chemotherapy-induced skeletal muscle loss and the response to NAC.

## 5. Conclusions

We observed that skeletal muscle loss, especially treatment-induced skeletal muscle loss, is associated with the response to NAC in patients with locally advanced breast cancer. The cancer stage is also significantly associated with a poor response to NAC. In the era of precision medicine, an accurate measurement of muscle mass will help guide treatment to achieve optimal clinical outcomes. Muscle-strengthening exercises, nutritional support, and pharmacologic intervention would be helpful in delaying muscle degradation and consequently enhancing the response to NAC in patients with breast cancer.

## Figures and Tables

**Figure 1 cancers-13-01806-f001:**
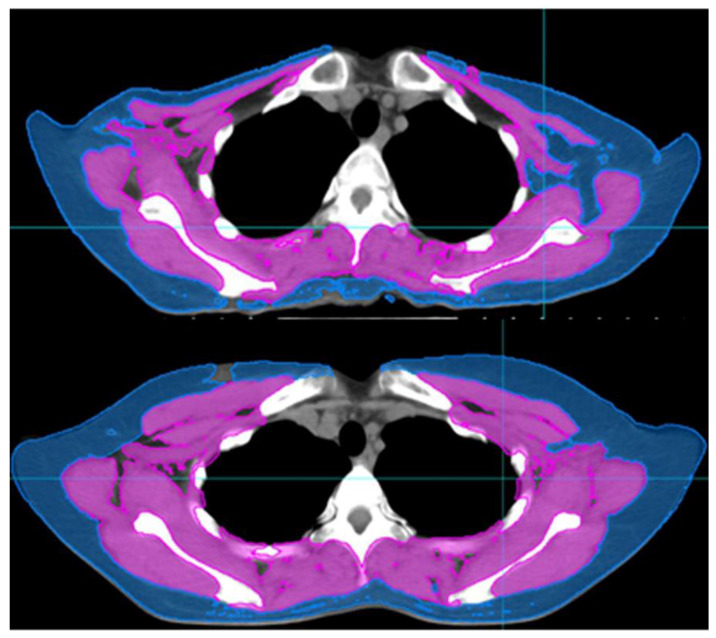
Computed tomography images of patients with skeletal muscle loss (**upper**) and without skeletal muscle loss (**lower**). skeletal muscle index of upper image: 4.88, skeletal muscle index of lower image: 9.83.

**Figure 2 cancers-13-01806-f002:**
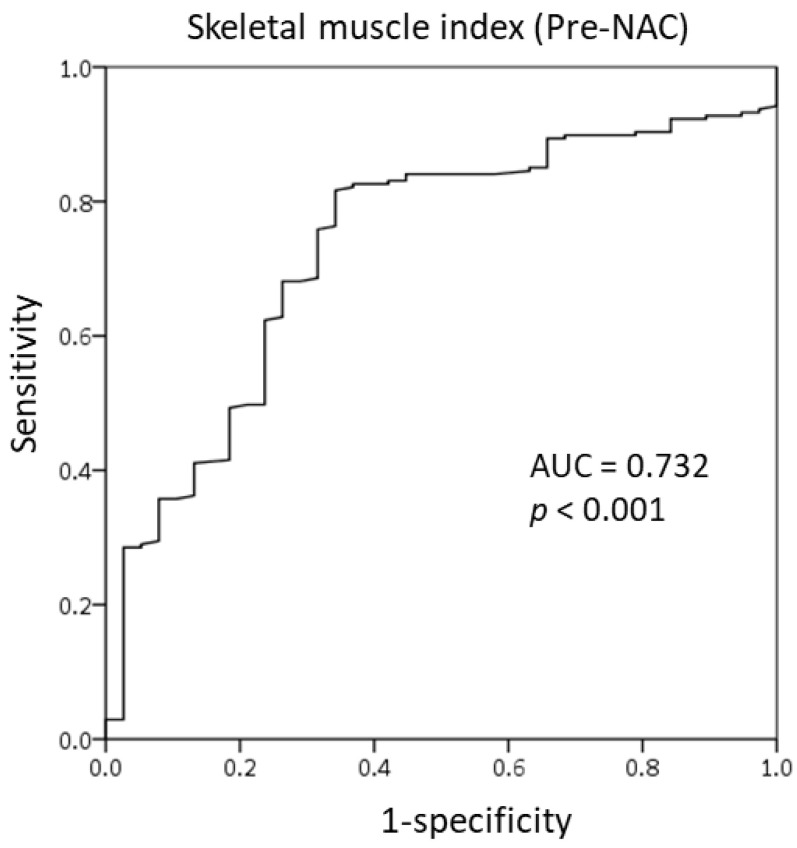
Receiver operating characteristic curve of the skeletal muscle index at the T4 level prior to the initiation of neoadjuvant chemotherapy.

**Figure 3 cancers-13-01806-f003:**
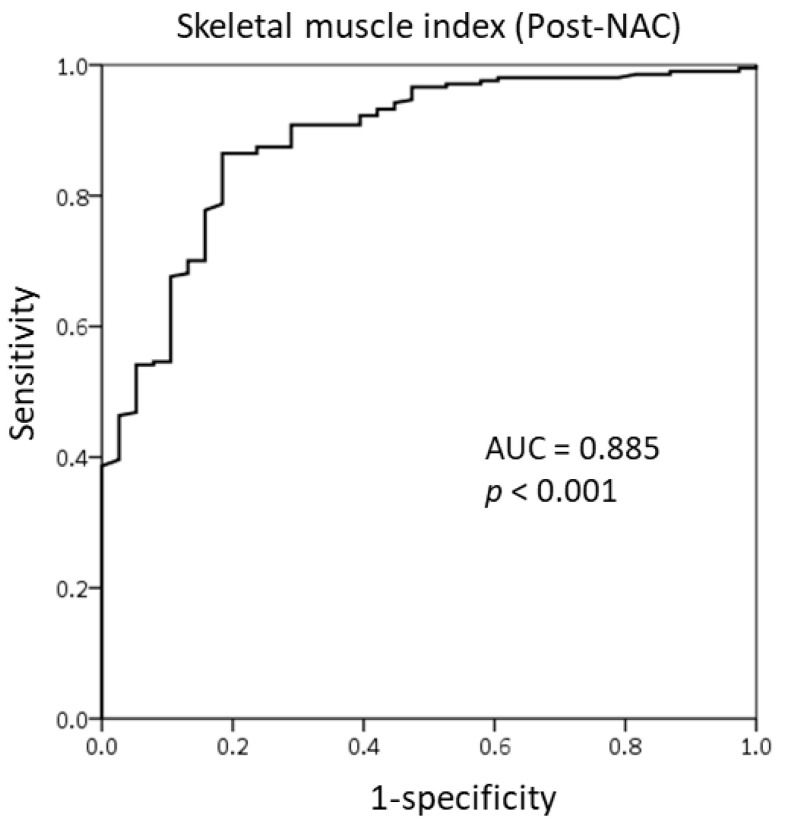
Receiver operating characteristic curve of the skeletal muscle index at the T4 level after the completion of neoadjuvant chemotherapy.

**Table 1 cancers-13-01806-t001:** Patient characteristics.

Characteristics	Number	%
Age		
Median (IQR)	48 (42–54)	
Hypertension		
No	215	87.40%
Yes	31	12.60%
Diabetes mellitus		
No	229	93.10%
Yes	17	6.90%
Exercise		
No	175	71.10%
Yes	71	28.90%
Weight change		
No change	211	85.80%
Decrease	14	5.70%
Increase	21	8.50%
Smoking status		
Non-smoker	241	98.00%
Current smoker	4	1.60%
Ex-smoker	1	0.40%
Drinking status		
Non-drinker	219	88.90%
Current drinker	21	8.60%
Ex-drinker	6	2.50%
Skeletal muscle mass_pre-NAC		
Skeletal muscle loss	30	12.20%
Non-skeletal muscle loss	216	87.80%
Skeletal muscle mass_post-NAC		
Skeletal muscle loss	33	13.50%
Non-skeletal muscle loss	213	86.50%
Change in skeletal muscle mass		
Skeletal muscle loss → Skeletal muscle loss	18	7.30%
Skeletal muscle loss → Non-skeletal muscle loss	12	4.90%
Non-skeletal muscle loss → Skeletal muscle loss	15	6.10%
Non-skeletal muscle loss → Non-skeletal muscle loss	201	81.70%
Body mass index		
Underweight (<18.5)	8	3.30%
Normal (18.5–23)	128	52.00%
Overweight (23–25)	43	17.50%
Obese (≥25)	67	27.20%
Hemoglobin		
Median (IQR)	13.2 (12.5–13.9)	
Platelets		
Median (IQR)	268 (229–312)	
Albumin		
Median (IQR)	4.5 (4.3–4.6)	
Protein		
Median (IQR)	7.4 (7.1–7.7)	

Abbreviations: IQR, Interquartile range; NAC, Neoadjuvant chemotherapy.

**Table 2 cancers-13-01806-t002:** Tumor characteristics.

Characteristics	Number	%
Pathology		
IDC	232	94.30%
ILC	5	2.05%
Mucinous	5	2.05%
Others	4	1.60%
Clinical T stage		
T1	46	18.70%
T2	139	56.50%
T3	34	13.80%
T4	27	11.00%
Clinical N stage		
N0	58	23.60%
N1	98	39.80%
N2	49	19.90%
N3	41	16.70%
Stage		
I	9	3.70%
II	123	50.00%
III	114	46.30%
Operation		
Breast-conserving surgery	129	52.40%
Modified radical mastectomy	117	47.60%
RT modality		
3D CRT	57	23.20%
IMRT	189	76.80%
Dose scheme		
180 cGy per fraction	226	91.90%
200 cGy per fraction	8	3.20%
267 cGy per fraction	12	4.90%
ER		
Negative	99	40.20%
Positive	147	59.80%
PR		
Negative	171	69.50%
Positive	75	30.50%
HER2		
Negative	170	69.10%
Positive	76	30.90%
Ki-67		
<15	112	45.50%
≥15	78	31.70%
Unknown	56	22.80%
Chemotherapy regimen		
Adriamycin based	5	2.00%
Adriamycin and taxol based	161	65.40%
Taxol based	8	3.30%
Trastuzumab based	72	29.30%
Selective estrogen receptor modulator		
No	143	58.10%
Yes	103	41.90%
Aromatase inhibitor		
No	220	89.40%
Yes	26	10.60%

Abbreviations: IDC, Invasive ductal carcinoma; ILC, Invasive lobular carcinoma; T stage, Tumor stage; N stage, Node stage; RT, Radiotherapy; 3D CRT, 3-dimensional conformal radiotherapy; IMRT, Intensity-modulated radiation therapy; ER, Estrogen receptor; PR, Progesterone receptor, HER2, Human epidermal growth factor receptor 2.

**Table 3 cancers-13-01806-t003:** Comparison of patient characteristics according to the response to NAC.

	Responder Group (*n* = 208)		Non-Responder Group (*n* = 38)		
Characteristics	Number	%	Number	%	*p*-Value
Age					
Median (IQR)	48		47		
Hypertension					0.911
No	182	87.50%	33	86.80%	
Yes	26	12.50%	5	13.20%	
Diabetes mellitus					0.663
No	193	92.80%	36	94.70%	
Yes	15	7.20%	2	5.30%	
Exercise					0.706
No	147	70.70%	28	73.70%	
Yes	61	29.30%	10	26.30%	
Weight change					0.811
No change	179	86.00%	32	84.20%	
Decrease	11	5.30%	3	7.90%	
Increase	18	8.70%	3	7.90%	
Smoking status					0.796
Non-smoker	204	98.10%	37	97.40%	
Current smoker	3	1.50%	1	2.60%	
Ex-smoker	1	0.50%	0	0.00%	
Drinking status					0.896
Non-drinker	186	89.30%	33	86.80%	
Current drinker	17	8.30%	4	10.50%	
Ex-drinker	5	2.40%	1	2.70%	
Skeletal muscle mass_pre-NAC					0.019
Skeletal muscle loss	21	10.10%	9	23.70%	
Non- Skeletal muscle loss	187	89.90%	29	76.30%	
Skeletal muscle mass post-NAC					<0.001
Skeletal muscle loss	12	5.80%	21	55.30%	
Non- Skeletal muscle loss	196	94.20%	17	44.70%	
Change in skeletal muscle mass					<0.001
Skeletal muscle loss → Skeletal muscle loss	9	4.30%	9	23.70%	
Skeletal muscle loss → Non-skeletal muscle loss	12	5.80%	0	0.00%	
Non-skeletal muscle loss → Skeletal muscle loss	3	1.40%	12	31.60%	
Non-skeletal muscle loss → Non-skeletal muscle loss	184	88.50%	17	44.70%	
Body mass index					0.903
Underweight	7	3.40%	1	2.60%	
Normal	107	51.40%	21	55.30%	
Overweight	36	17.30%	7	18.40%	
Obese	58	27.90%	9	23.70%	
Hemoglobin					0.922
Median (IQR)	13.2		13.4		
Platelets					0.051
Median (IQR)	264.5		292.5		
Albumin					0.125
Median (IQR)	4.5		4.6		
Protein					0.937
Median (IQR)	7.4		7.45		

Abbreviations: IQR, Interquartile range; NAC, Neoadjuvant chemotherapy.

**Table 4 cancers-13-01806-t004:** Comparison of tumor characteristics according to the response to neoadjuvant chemotherapy.

	Responder Group (*n* = 208)		Non-Responder Group (*n* = 38)		
Characteristics	Number	%	Number	%	*p*-Value
Pathology					0.001
IDC	201	96.60%	31	81.60%	
ILC	2	1.00%	3	7.90%	
Mucinous	2	1.00%	3	7.90%	
Others	3	1.40%	1	2.60%	
T stage					0.116
T1	42	20.20%	4	10.50%	
T2	120	57.70%	19	50.00%	
T3	26	12.50%	8	21.10%	
T4	20	9.60%	7	18.40%	
N stage					0.076
N0	52	25.00%	6	15.80%	
N1	87	41.80%	11	28.90%	
N2	37	17.80%	12	31.60%	
N3	32	15.40%	9	23.70%	
Stage					0.003
I	9	4.30%	0	0.00%	
II	112	53.90%	11	28.90%	
III	87	41.80%	27	71.10%	
Operation					<0.001
Breast-conserving surgery	120	57.70%	9	23.70%	
Modified radical mastectomy	88	42.30%	29	76.30%	
RT modality					0.241
3D CRT	51	24.50%	6	15.80%	
IMRT	157	75.50%	32	84.20%	
Dose scheme					0.354
180 cGy per fraction	189	90.90%	37	97.40%	
200 cGy per fraction	8	3.80%	0	0.00%	
267 cGy per fraction	11	5.30%	1	2.60%	
ER					0.236
Negative	87	41.80%	12	31.60%	
Positive	121	58.20%	26	68.40%	
PR					0.874
Negative	145	69.70%	26	68.40%	
Positive	63	30.30%	12	31.60%	
HER2					0.296
Negative	141	67.80%	29	76.30%	
Positive	67	32.20%	9	23.70%	
Ki-67					0.001
<15	92	44.20%	20	52.60%	
≥15	60	28.80%	18	47.40%	
Unknown	56	27.00%	0	0.00%	
Chemotherapy regimen					<0.001
Adriamycin based	1	0.50%	4	10.50%	
Adriamycin and taxol based	138	66.30%	23	60.50%	
Taxol based	5	2.40%	3	7.90%	
Trastuzumab based	64	30.80%	8	21.10%	
Selective estrogen receptor modulator					0.162
No	117	56.25%	26	68.40%	
Yes	91	43.75%	12	31.60%	
Aromatase inhibitor					0.560
No	185	88.90%	35	92.10%	
Yes	23	11.10%	3	7.90%	

Abbreviations: IDC, Invasive ductal carcinoma; ILC, Invasive lobular carcinoma; T stage, Tumor stage; N stage, Node stage; RT, Radiotherapy; 3D CRT, 3-dimensional conformal radiotherapy; IMRT, Intensity-modulated radiation therapy; ER, Estrogen receptor; PR, Progesterone receptor, HER2, Human epidermal growth factor receptor 2.

**Table 5 cancers-13-01806-t005:** Univariate and multivariate analyses of the response to NAC.

	Univariate Analysis			Multivariate Analysis		
Variables	HR	95% CI	*p*-Value	HR	95% CI	*p*-Value
Age (<48 vs. ≥48 years)	1.322	0.66–2.64	0.430			
Smoking status			0.877			
Non-smoker vs. current smoker	0.550	0.06–5.43	0.608			
Non-smoker vs. ex-smoker	NA	NA	NA			
Drinking status			0.896			
Non-drinker vs. current drinker	0.762	0.24–2.41	0.644			
Non-drinker vs. ex-drinker	0.897	0.10–7.92	0.922			
Pre-NAC skeletal muscle mass (Skeletal muscle loss vs. non-skeletal muscle loss)	2.749	1.15–6.58	0.023	0.193	0.04–0.94	0.042
Post-NAC skeletal muscle mass (Skeletal muscle loss vs. non-skeletal muscle loss)	20.074	8.45–47.69	<0.001	64.566	15.13–275.58	<0.001
Change in skeletal muscle mass ^†^			<0.001			
Group 1 vs. 2	N/A	N/A	0.999			
Group 1 vs. 3	0.250	0.52–1.20	0.083			
Group 1 vs. 4	10.824	3.79–30.90	<0.001			
Body mass index			0.904			
Underweight vs. Obese	1.086	0.12–9.90	0.942			
Normal vs. Obese	0.739	0.32–1.72	0.483			
Overweight vs. Obese	0.798	0.27–2.33	0.680			
ER (negative vs. positive)	0.723	0.35–1.49	0.381			
PR (negative vs. positive)	1.028	0.51–2.07	0.938			
HER2 (negative vs. positive)	1.465	0.66–3.27	0.352			
Ki-67 (<15 vs. ≥15)	0.725	0.35–1.48	0.377			
Pathology			0.009			0.056
IDC vs. ILC	0.103	0.02–0.64	0.015	1.209	0.09–16.90	0.888
IDC vs. mucinous	0.103	0.02–0.64	0.015	0.067	0.01–0.47	0.006
IDC vs. others	0.463	0.05–4.59	0.510	0.640	0.03–12.71	0.770
T stage			0.131			
T1 vs. T2	0.602	0.19–1.87	0.380			
T1 vs. T3	0.310	0.09–1.13	0.076			
T1 vs. T4	0.272	0.07–1.04	0.057			
N stage			0.086			
N0 vs. N1	0.913	0.32–2.61	0.865			
N0 vs. N2	0.356	0.12–1.03	0.058			
N0 vs. N3	0.410	0.13–1.26	0.120			
Stage (stage I/II vs. stage III)	0.293	0.14–0.62	0.001	0.276	0.11–0.71	0.007

Abbreviations: NAC, Neoadjuvant chemotherapy; HR, Hazard ratio; CI, Confidence interval; NA, Not applicable; ER, Estrogen receptor; PR, Progesterone receptor, HER2, Human epidermal growth factor receptor 2; IDC, Invasive ductal carcinoma; ILC, Invasive lobular carcinoma; T stage, Tumor stage; N stage, Node stage. ^†^ Group 1: Patients with skeletal muscle loss in both pre-NAC and post-NAC, Group 2: Patients who had skeletal muscle loss in pre-NAC but changed to non-skeletal muscle loss in post-NAC, Group 3: Patients who changed from non-skeletal muscle loss in pre-NAC to skeletal muscle loss in post-NAC, Group 4: Patients with non-skeletal muscle loss in both pre-NAC and post-NAC.

## Data Availability

The data presented in this study are available on request from the corresponding author. The data are not publicly available due to privacy.

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
