# Peer review of "Association between Skeletal Muscle Loss and the Response to Neoadjuvant Chemotherapy for Breast Cancer"

_cancers, 2021, doi:10.3390/cancers13081806_

Round 1

Reviewer 1 Report

Lee et al. evaluated the impact of sarcopenia on responsiveness of neoadjuvant chemotherapy in breast cancer. This is an interesting paper, indicating that the prediction of response of neoadjuvant chemotherapy could be helpful to guide the treatment direction. There are several limitations to be addressed to further improve the manuscript.

Author Response

Thank you for the time and effort of editors and reviewers put into our manuscript. We revised the manuscript based on comments of reviewer and made a point by point response. The changes in the manuscript are highlighted in yellow. We are pleased to additional comments and suggestions. 

Thank you very much about the comment. I supplement the limitation of this study in discussion session.

Revised manuscript> There are several limitations to this study. Due to the retrospective nature, the causal relationships between the variables and sarcopenia could not be determined. The results of this study indicated that the muscle wasting at the end of neoadjuvant treatment is more associated with the response to treatment. The muscle wasting during the treatment could be influenced by the tumor aggressiveness. Due to this reason, the careful attention should be paid in interpreting the results.

Reviewer 2 Report

Very interesting manuscript showing how the presence of reduced muscle mass could represent a predictor of the lack of response to neoadjuvant chemotherapy and radiotherapy for breast cancer. The topic is worth of investigation and the results could be very important for clinical practice. There are however some issues needing clarification. 

1) The main issue regards the definition of sarcopenia. In geriatrics, sarcopenia is defined as an age-related loss of muscle mass and function. Despite the concept of sarcopenia has been operationalized in older invidivuals, it can be applied also to younger subjects, especially if suffering from chronic diseases or disabled. In this manuscript, the authors measured only muscle mass, not muscle function, and yet persistently refer to reduced muscle mass as sarcopenia. This definition is not completely correct: the authors should have performed also some tests of muscle function, such as the measurement of handgrip strength. Moreover, the definition of sarcopenia used by authors is not compliant with any of the international consensuses on sarcopenia (for example, EWGSOP, FNIH, Asian Working Group on Sarcopenia). Notably, the Asian Working Group on Sarcopenia defined this clinical entity as "age-related loss of muscle mass, plus low muscle strength and/or low physical performance". I think that the manuscript should be revised in accordance with these concepts. Maybe the authors could substitute the concept of "sarcopenia" with the concept of "muscle wasting", which is more neutral and does not imply an age-related entity causing loss of physical function and disability. 

2) The abstract is clear, but it should also contain information on the age of participants, because sarcopenia/muscle wasting are generally considered age-related conditions. 

3) Did all the participants receive radiotherapy as part of the neoadjuvant cancer treatment? If so, was it standardized or not? The methods section should be rephrased because these procedures are not clear. 

4) The criteria for classification of patients as responders or non-responders should be presented more clearly, because they represent the main endpoint of the study. 

5) The study could be underpowered, because only a limited number of participants was classified as non-responders. Was any power calculation performed? 

6) From a clinical perspective, the association between baseline muscle wasting and failure to respond to treatment is much more important than the association between muscle wasting and response to treatment at the end of the study. The authors should bear in mind this concept when presenting results and discussing them. However, the neoadjuvant treatment regimens were very heterogeneous and of course influenced by the disease staging and baseline prognostic factors. Thus, the association between muscle wasting at the end of treatment and treatment failure could be influenced by tumor aggressiveness and drugs administered. The importance of assessing muscle mass before starting a neoadjuvant treatment for breast cancer is anyway well supported by the results. 

Author Response

Thank you for the time and effort of editors and reviewers put into our manuscript. We revised the manuscript based on comments of reviewer and made a point by point response. The changes in the manuscript are highlighted in yellow. We are pleased to additional comments and suggestions. 

1) Response> I really appreciate to your critical comments. I understand that there was confusing definition of sarcopenia. As you pointed out, we did not evaluate the muscle function. We changed the expression “sarcopenia” into “skeletal muscle loss”

2) Response> Thank you for your valuable comments. I added the information on the age of participants in the abstract.

3) Response> All the patients underwent radiotherapy as part of the adjuvant treatment, not neoadjuvant treatment. I am sorry for the unclear expression on the method section. We added the explanation on the page 2, line 66-69.

Revised manuscript> Every patient in this study received neoadjuvant chemotherapy, operation and adjuvant radiotherapy. The adjuvant radiotherapy was performed in patients with tumor stage III/IV or extensively positive lymph node initially among the patients who underwent total mastectomy.

4) Response> I am sorry for the unclear expression. I added the explanation about the classification of patients of responder and non-responder. We assessed the treatment response based on the image study after the neoadjuvant chemotherapy. I added the explanation in the revised manuscript in page 3 line 102 - 105.

Revised manuscript> The treatment response was evaluated based on the image study of the patients. The MRI (Magnetic Resonance Imaging) was used for the evaluation before and after the NAC. Patients showing either a complete response or partial response in the MRI performed after the NAC were classified into the responder group, while patients with either stable disease or progressive disease were categorized into the non-responder group.

5) Response> Thank you for your valuable comments. For your good advice, we sought advice from medical statistics expert. Based on the counseling from statistics expert, the skeletal muscle loss was significant factor associated with response of chemotherapy despite the small number of non-responder group. Even after performing the power calculation, the result will not be changed. I really appreciate your critical comments.

6) Response> Thank you for the critical suggested comment. I fully understand the point you suggested. However, as this study is retrospective study, we cannot fully understand the causal relationships. I supplemented about this limitation in the discussion session in page 15.

Revised manuscript> There are several limitations to this study. Due to the retrospective nature, the causal relationships between the variables and sarcopenia could not be determined. The results of this study indicated that the muscle wasting at the end of neoadjuvant treatment is more associated with the response to treatment. The muscle wasting during the treatment could be influenced by the tumor aggressiveness. Due to this reason, the careful attention should be paid in interpreting the results.

Reviewer 3 Report

Association between sarcopenia and the response to neoadjuvant chemotherapy for breast cancer

Interesting paper.  I would add BMI to the data. The electronic medical record generally calculates this for clinicians. If the BMI is not calculated by the electronic platform the data should be in the medical records.

Were patients in the study assessed for risk of or presence of malnutrition either before NAC or after NAC? Malnutrition and sarcopenia are often concurrent.

Author Response

Thank you for the time and effort of editors and reviewers put into our manuscript. We revised the manuscript based on comments of reviewer and made a point by point response. The changes in the manuscript are highlighted in yellow. We are pleased to additional comments and suggestions. 

Response> Thank you for the critical comments. I added the BMI data on this study. Even though the BMI data did not show the significant association of treatment response. I stratified the patients into 4 groups according to BMI; underweight (BMI<18.5), normal (BMI 18.5 – 22.9), overweight (BMI 23.0 – 24.9), obese (BMI≥25.0). I attached the data regarding BMI in table 1 and table 5. Thank you for your critical comments.

Response> Serum albumin is one of the marker representing the nutritional status of patients. We assessed the albumin level both before NAC and after NAC. None of the patients showed hypoalbuminemia at both before NAC and after NAC. Also, there was no correlation between albumin level and muscle mass index.

Skeletal mass index_Pre_NAC

Skeletal mass index_Post_NAC

Albumin_Pre_NAC

Correlation coefficient

-0.072

-0.101

Significant

0.259

0.116

Albumin_Post_NAC

Correlation coefficient

-0.098

-0.092

Significant

0.127

0.152

Every patient answered the questionnaire about their physical status before hospitalization. The questions about the change of the oral intake is included. In Most of the patients answered no change of the oral intake, but 9 patients answered the decrease of their oral intake. 7 patients showed decrease of the oral intake among the non-skeletal muscle loss group (3.30%) while only 2 patients answered decrease of the oral intake in skeletal muscle loss (6.10%).

Skeletal muscle mass index (Post-NAC)

Skeletal muscle loss

Non-skeletal muscle loss

N

%

N

%

p-value

Oral intake

No change

29

87.90%

206

96.70%

0.001

Decrease

2

6.10%

7

3.30%

Increase

2

6.10%

0

0.00%

Although the loss of oral intake does not clearly reflect the patient's malnutrition status, it may be one of the risk of malnutrition.

Round 2

Reviewer 1 Report

The authors put an effort in revising their manuscript and addressing issues raised previously.

The manuscript has been improved. Given the nature of the work, several aspects still remain speculative. The authors’ future study should clarify these aspects.

Reviewer 2 Report

The authors have adequately responded to all my previous comments. I have no further comments.